# Low-Pressure Laparoscopy Using the AirSeal System versus Standard Insufflation in Early-Stage Endometrial Cancer: A Multicenter, Retrospective Study (ARIEL Study)

**DOI:** 10.3390/healthcare10030531

**Published:** 2022-03-14

**Authors:** Alessandro Buda, Giampaolo Di Martino, Martina Borghese, Stefano Restaino, Alessandra Surace, Andrea Puppo, Sara Paracchini, Debora Ferrari, Stefania Perotto, Antonia Novelli, Elena De Ponti, Chiara Borghi, Francesco Fanfani, Robert Fruscio

**Affiliations:** 1Division of Gynecologic Oncology, Michele e Pietro Ferrero Hospital, 12060 Verduno, Italy; alessandra.sur@gmail.com (A.S.); saraparacchini89@gmail.com (S.P.); stperotto@aslcn2.it (S.P.); borghi.chr@gmail.com (C.B.); 2Clinic of Gynecology and Obstetrics, San Gerardo Hospital, University of Milano-Bicocca, 20900 Monza, Italy; giamp.dima@gmail.com (G.D.M.); deboraferrari6@gmail.com (D.F.); robert.fruscio@unimib.it (R.F.); 3Clinic of Obstetrics and Gynecology, Santa Croce e Carle Hospital, 12100 Cuneo, Italy; martina.borghese1989@gmail.com (M.B.); puppoand@hotmail.com (A.P.); antonianovelli@hotmail.it (A.N.); 4Division of Obstetrics and Gynecology, University Hospital of Udine, Azienda Sanitaria Universitaria Friuli Centrale, 33100 Udine, Italy; restaino.stefano@gmail.com; 5Division of Medical Physics, ASST Monza, San Gerardo Hospital, University of Milan-Bicocca, 33100 Monza, Italy; elena.deponti@unimib.it; 6Dipartimento della Salute della Donna, del Bambino e di Sanità Pubblica, Fondazione Policlinico Universitario A. Gemelli IRCCS, 00168 Rome, Italy; francesco.fanfani74@gmail.com; 7Dipartimento Scienze della Vita e Sanità Pubblica, Università Cattolica del Sacro Cuore, 00168 Rome, Italy

**Keywords:** laparoscopy, endometrial cancer, low-pressure insufflation, postoperative pain

## Abstract

The aim of our study was to evaluate the benefits of a low-pressure insufflation system (AirSeal) vs. a standard insufflation system in terms of anesthesiologists’ parameters and postoperative pain in patients undergoing laparoscopic surgery for early-stage endometrial cancer. This retrospective study involved five tertiary centers and included 152 patients with apparent early-stage disease who underwent laparoscopic surgical staging with either the low-pressure AirSeal system (8–10 mmHg, n = 84) or standard laparoscopic insufflation (10–12 mmHg, n = 68). All the intraoperative anesthesia variables evaluated (systolic blood pressure, end-tidal CO2, peak airway pressure) were significantly lower in the AirSeal group. We recorded a statistically significant difference between the two groups in the median NRS scores for global pain recorded at 4, 8, and 24 h, and for overall shoulder pain after surgery. Significantly more women in the AirSeal group were also discharged on day one compared to the standard group. All such results were confirmed when analyzing the subgroup of women with a BMI >30 kg/m^2^. In conclusion, according to our preliminary study, low-pressure laparoscopy represents a valid alternative to standard laparoscopy and could facilitate the development of outpatient surgery.

## 1. Introduction

Over the last two decades, laparoscopy has become the favorite approach for the surgical treatment of apparent early-stage endometrial cancer worldwide [1,2,3,4]. Moreover, minimally invasive surgery is particularly indicated for obese patients, since it seems to prevent the majority of postoperative complications [5]. However, endometrial cancer patients often have several comorbidities, and the setup of the optimal intra-abdominal pressure (IAP) is demanding. On the one hand, high-pressure pneumoperitoneum enables a satisfactory working space and the optimal visualization of the surgical field, shortening operative time and blood loss and ultimately improving surgical performance. On the other hand, however, several complications are related to the use of CO2, particularly in patients with compromised lung and heart function [6]. Increased IAP that is induced by the pneumoperitoneum can lead to several important hemodynamic alterations, such as to the acid–basic balance, and can cause significant postoperative shoulder pain [7,8,9]. Innovative, valveless low-pressure trocars such as the AirSeal system (AirSeal^®^, ConMed, Utica, NY, USA), have been introduced and appear to be a valid alternative to the standard insufflation system based on preliminary studies. This system provides a more stable pneumoperitoneum by responding to the slightest changes in IAP and is associated with a reduced rate of CO2 use, absorption, and elimination [10,11]. Furthermore, CO2 use is demonstrated to be significantly decreased using such a system, resulting in a potential reduction in CO2-related complications; this is of particular benefit for patients with impaired cardiopulmonary function [11]. In randomized experiments in urology, the system improved visualization and reduced cardiopulmonary damage compared to standard insufflation [12,13,14]. The AirSeal platform has been evaluated in general gynecologic surgery, mainly in cases of benign disease [15,16]. However, data from a gynecologic oncologic setting are still lacking. Hence, the aim of this retrospective pilot study was to evaluate the role of low-pressure pneumoperitoneum (8–10 mmHg) with the AirSeal system in patients with apparent early-stage endometrial cancer. In addition, we explored the impact of low-pressure insufflation on intraoperative anesthesiology parameters and postoperative patients’ pain compared to the standard pressure insufflation system.

## 2. Materials and Methods

This was a multicenter, two-arm, retrospective pilot study conducted at the Michele e Pietro Ferrero Hospital, Verduno (Italy); San Gerardo Hospital, University of Milano-Bicocca, Monza (Italy); Ospedale Santa Croce e Carle, Cuneo (Italy); Division of Obstetrics and Gynecology, University Hospital of Udine; and at the Fondazione Policlinico Universitario A. Gemelli, IRCCS, Rome (Italy). The ethics committee of the principal investigator has declared that it was not necessary to obtain informed consent from patients for this retrospective study specifically, since all patients had given informed consent before surgery.

One hundred and fifty-two women with apparent early-stage endometrial cancer who had undergone laparoscopic surgical staging with the AirSeal System between January 2019 and December 2021 were retrospectively included in this study and were compared to a subgroup of women operated with the standard insufflation system.

The primary objective was to compare and evaluate the impact of the low-insufflation system in terms of anesthesiology parameters and postoperative shoulder pain at 4, 8 and 24 h after surgery. Intra-operatively, the following were evaluated: systolic blood pressure, end-tidal CO2, peak airway pressure, volume of CO2 used, duration of surgery, and blood loss. Pain was evaluated with the Numeric Rating Scale (NRS) with a range between 0 and 10.

Secondarily, we investigated the impact of low-pressure insufflation in the subgroup of women with a BMI > 30 mg/m^2^. Considering the retrospective nature of the study, which did not involve any direct patient contact or diagnostic or therapeutic intervention, a waiver of informed consent and a waiver of authorization were requested. All patient information was guaranteed to be confidential. The study was conducted according to the guidelines of the Declaration of Helsinki.

### 2.1. Operative Procedure

After the induction of standard general anesthesia, the induction of pneumoperitoneum was achieved by using either the Veres needle technique or the open technique at the standard flow of 12 mmHg. After primary port placement, the intra-abdominal pressure was decreased to 8–10 mmHg in the AirSeal system group, while it was maintained at 12–14 mmHg in the standard insufflation group.

The AirSeal insufflation system, once activated, works with three essential components, the iFS AirSeal control, the Tri-Lumen Filtered Tube Set, and the Access Port, which together allow the creation of a stable pneumoperitoneum and of continuous smoke evacuation. The access port is available in two sizes: 5 or 10 mm. The 5 mm port was usually placed on the right side of the iliac quadrant, whereas the 10 mm port was placed on the camera site, i.e., on the umbilicus scar. In order to prevent smoke evacuation from impairing visualization, the 5 mm trocar was placed away from the camera. Two more accessory 5 mm diameter trocars that do not require fascial closure were placed, as shown in Figure 1.

The general anesthesia induction protocol included Propofol as a hypnotic, Fentanyl or Remifentanyl as an analgesic, and Rocuronium as a muscle relaxant, followed by intubation, and maintenance with Desflurane or Sevoflurane and Remifentanil. Nausea and vomiting prevention were obtained with Dexamethasone administered at 4 or 8 milligrams intravenously.

In the absence of contraindications, postoperative pain was controlled with the intravenous administration of Paracetamol at 1 g and Ketoprofene at 160 milligrams (Artrosilene) 3 times daily. In the presence of a Numeric Rating Scale of pain (NRS) higher than 4, Ketoprofene 160 mg was given intravenously and, eventually, subcutaneous morphine 10 mg was given if NRS was still higher than 4 after administration of Ketoprofene.

### 2.2. Statistical Analysis

Descriptive statistics are presented as frequencies and proportions for categorical variables and mean/median with a standard deviation/interquartile range for continuous variables. These descriptive characteristics are reported in tables so that readers can inspect the distribution of these variables in the two study groups (ultrastaging protocol A vs. B) and for the lymph node status (positive node disease vs. negative node disease). Continuous variables were compared using Wilcoxon rank-sum tests. Proportions were compared using Chi-Square tests or Fisher’s exact tests. All 2-sided *p*-values with *p* < 0.05 were considered statistically significant. We performed a multivariate analysis to adjust for potential confounding factors. We used logistic regression to perform the analysis because of the binary endpoint. Stata software 9.0 (Stata Corporation, College Station, TX, USA) was used for performing the statistical analysis.

## 3. Results

One hundred and fifty-two patients were included in this retrospective study. A total of 84 patients (55.3%) underwent laparoscopic surgery with the AirSeal system at 8–10 mmHg, whereas 68 (44.7%) were treated with conventional laparoscopic surgery and the standard insufflation system. The baseline characteristics of the study population are shown in Table 1.

Surgical procedures, including simple hysterectomy, bilateral salpingo-oophorectomy and sentinel node biopsy, were performed with a traditional laparoscopic tower. The median age of patients was 63 years (IQR 52–73) for the AirSeal group, and 65 years (IQR 54–75) for the standard group. The median BMI was 30.0 kg/m^2^ (range 23.7–36.4) and 27.9 kg/m^2^ (range 22.6–33.2) for the AirSeal and the standard group, respectively. No differences in terms of the presence of previous surgery, smoking habits, pulmonary disease and ASA score were observed between the groups. Surgical outcomes such as duration of surgery, blood loss, and number of lymph nodes removed were similar between the two groups.

Median length of hospital stay was 2 days in both groups (*p* = 0.224). However, patients in the AirSeal group recovered significantly faster compared to the standard group. In the AirSeal group, 98% (80/81) of women were discharged within 2 days, compared to 75% (51/68) of patients in the standard group (*p* < 0.0001). The complication rate did not differ between the two groups.

### 3.1. Anesthesia Parameter Results

All the intraoperative anesthesia variables evaluated (systolic blood pressure, end-tidal CO2, peak airway pressure) were significantly lower in the AirSeal group (Table 2). Furthermore, the median volume of CO2 consumed during surgery was 35 L (IQR 30–36) in the AirSeal group, and 50 L (IQR 40–50) in the standard group (*p* < 0.0001). In the subgroup of 67 women with a BMI greater than 30 mg/m^2^, the differences were maintained (Table 3).

### 3.2. Pain Control Results

We recorded a statistically significant difference in the median NRS scores for global pain recorded at 4, 8, and 24 h and for overall shoulder pain after surgery between the two groups (Table 2). The median NRS score at 4 h was 1.4 (IQR range 0.4–2.4) for the AirSeal group and 1.7 (IQR range 0.9–2.5) for the standard group (*p* = 0.023). This difference was maintained at 8 and 24 h after surgery, when the median reported global pain was 0.8 (IQR range 0–1.6) and 0.2 (IQR range 0–0.8), respectively, for the AirSeal arm compared to 1.4 (IQR range 0.7–2.1) and 1.1 (IQR range 0.2–2.0) for the control arm (*p* < 0.0001). The administration of morphine was statistically different across the two groups, with 4.8% and 27.9% in the AirSeal and the standard group, respectively (*p* < 0.0001). The analysis of women with a BMI greater than 30 mg/m^2^ showed that the differences between the Airseal and standard groups were maintained (Table 3).

Figure 1 shows the NRS pain control score of the two groups at 4, 8 and 24 h after surgery.

## 4. Discussion

In this study, we found that in women with endometrial cancer, the use of low-pressure laparoscopy using the AirSeal valveless system significantly reduced the absorption of CO2 and improved the control of post-operative pain when compared to the standard insufflation system. The benefits of minimally invasive surgery (MIS) for endometrial cancer patients have been widely demonstrated in the last decade [2,17,18]. Women who underwent MIS surgery—traditional laparoscopy or robotic-assisted surgery—including hysterectomy and surgical staging, showed better operative and post-operative outcomes when compared with traditional open surgery. Obese women with endometrial cancer further benefited from the advantages of the MIS approach and experienced shorter hospitalization, less blood loss and less post-operative pain, a better quality of life and a lower risk of surgical morbidity including post-operative fever, post-operative ileus and wound infections. However, differently from healthy and fit patients without malignancies, patients with endometrial cancer frequently have many comorbidities, especially regarding obesity, a factor strongly associated with endometrial cancer, which entails anatomical and physiological changes that are a critical consideration for guaranteeing safe and successful surgery [19].

Pneumoperitoneum with CO2 increases the risks of respiratory, cardiovascular and metabolic complications, mainly caused by the increase in the intra-abdominal pressure that triggers the increase in median systolic pressure. In obese women, this phenomenon is accentuated and with the activation of the kidney-angiotensin renin system, a further increase in blood pressure can occur. Furthermore, elevation of the intra-abdominal pressure with pneumoperitoneum causes an upward elevation of the diaphragm, resulting in increased intra-thoracic pressure, thus decreasing pulmonary compliance and augmenting peak airway pressure, which results in an increased risk of barotrauma. In this study, the use of low-pressure insufflation with the AirSeal system allowed for a significant limitation of the increase in et-CO2, peak airway pressure and maximum systolic blood pressure during surgery. The overall consumption of CO2 was also reduced with AirSeal (35 L, IQR range 30–26) compared with traditional insufflation (50 L, IQR 40–50; *p* = 0.0001).

In this preliminary study, the improvement in terms of anesthesiology results was also maintained in the subgroup of women with a BMI greater than 30 kg/m^2^. Obesity increases the risk of surgical morbidity in women with endometrial cancer and minimally invasive surgery is recommended in the presence of co-morbidities to reduce the risk of post-operative complications [5].

Finally, the adoption of a valveless system also has the advantage of maintaining optimal exposure of the operative field even with a lower pressure of insufflation and allowing the continuous emission of surgical smoke.

The study by Herati et al., which included 51 patients undergoing laparoscopy for renal surgery, showed that the use of the AirSeal system significantly reduced CO2 absorption during laparoscopy when compared with the standard insufflation system [11]. Regarding gynecologic surgery, our results showed similar outcomes to the study of Sroussi et al. [6], which explored the feasibility of the use of AirSeal at 7 mmHg in a cohort of patients with a benign gynecological pathology. In our experience, the combination of low pressure and a valveless trocar reduced end tidal CO2, the incidence of hypercapnia and associated consequences such as emphysema. Peak airway pressure and systolic blood pressure were also significantly lower in the AirSeal group. These results seem important and should be considered for both anesthesiologists and surgeons when operating on older women, obese patients, or patients with impaired cardiopulmonary functions, in which these parameters can be reduced by using low pressure and the valveless AirSeal trocar.

These data seem to be reinforced further by our study, which included women with endometrial cancer involving longer surgical duration (median time: 26 and 30 min in Sroussi versus 120 min in our study). Moreover, as already shown by Sroussi et al., the pain control was lower overall in the AirSeal group compared to the standard insufflation group, in both the general population and the obese subgroup [6]. On the other hand, our study and those available in the literature are in contrast with the recent randomized study by Madueke-Laveaux et al. [16], which did not report statistically significant advantages of the valveless system in terms of CO2 absorption rates and post-operative pain control in women who underwent laparoscopic surgery for benign conditions. However, the study involved a healthy, younger, and non-obese population with a lower rate of associated co-morbidity when compared to other already published experiments [6,11]. Furthermore, the authors confirmed the improvement of the visualization of the operative field with AirSeal 15 min after the use of electrosurgery and at the time of colpotomy.

In our opinion, our results should further encourage the adoption of this technology in patients with a higher risk of complications such as obesity and older age with malignancies, as demonstrated by the LAP2 study, where laparoscopy did not increase with age and analysis showed that a higher percentage of patients had undergone the open approach [2]. Moreover, low-pressure laparoscopy showed a better overall control of post-operative pain at 4, 8 and 24 h as well as shoulder pain. These results were significantly lower than those obtained with the standard insufflation system. This hastened the patients’ discharge. In our study, 28% of women in the AirSeal group were discharged on day one compared to 6% in the standard group. Almost all patients were discharged within 2 days after surgery in the AirSeal group compared to 75% in the standard group (*p* = 0.0001). Even if outpatient surgery is not widespread in Italy, the development and increased adoption of low-pressure laparoscopy associated with 3 mm trocars [15], together with the pre-habilitation already proposed in the ERAS program [20], may further minimize surgical morbidity and can increase the application of outpatient laparoscopy for oncological surgery as well.

Our results have some drawbacks, including the retrospective design and the relatively small sample size, and should be interpreted with caution. However, the differences highlighted between groups are significant and further multicentric studies should evaluate and further confirm our preliminary results in the oncologic population.

In conclusion, the use of low-pressure laparoscopy should be encouraged mainly in the subgroup of obese women with endometrial cancer, as well as to promote the development of outpatient surgery. Low-pressure laparoscopy with the valveless platform in our experience represents a valid and innovative alternative to the standard insufflation system during minimally invasive surgery in gynecologic oncology.

## Figures and Tables

**Figure 1 healthcare-10-00531-f001:**
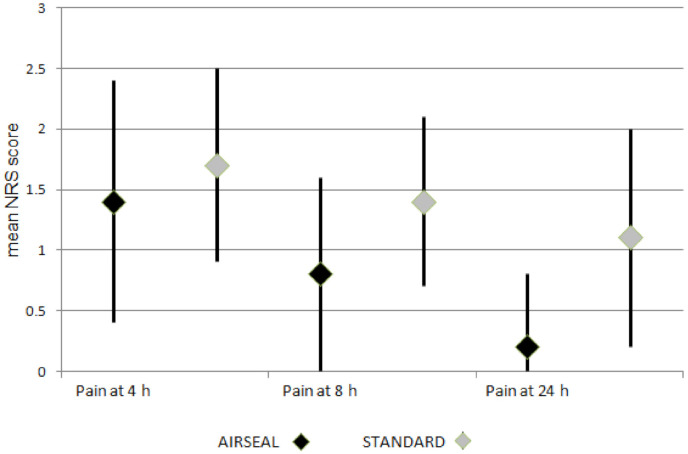
NRS pain control score of the two groups at 4, 8 and 24 h after surgery.

**Table 1 healthcare-10-00531-t001:** General characteristics of the study groups.

Variables	AirSeal (N = 84)	Standard (N = 68)	*p* Value
Age			
median (range)	63 (52–73)	65 (54–75)	
			0.275
BMI, kg/m^2^, median (range)	30.0 (23.7–36.4)	27.9 (22.6–33.2)	0.142
FIGO stage, n (%)			0.980
IA	54 (64.3)	44 (65.7)	
IB	14 (16.7)	12 (17.9)	
II	4 (4.8)	3 (4.5)	
III	12 (14.3)	8 (11.9)	
Atypical Hyperplasia	-	1	
Grading, n (%)			0.006
G1	48 (57.1)	20 (31.3)	
G2	24 (28.6)	27 (42.2)	
G3	12 (14.3)	17 (26.6)	
NA	0	4	
Histotype, n (%)			0.013
Complex hyperplasia	0	1 (1.5)	
Endometrioid	80 (95.2)	58 (85.3)	
Serous	-	6 (8.8)	
MMT & others	4 (4.8)	3 (4.4)	
LVSI, n (%)			0.121
Present	23 (27.4)	12 (17.7)	
Absent	59 (72.6)	56 (82.3)	
NA	2	0	
Previous surgery, n (%)	28 (33.3)	29 (42.7)	0.174
Smoke habit, n (%)	29 (34.5)	23 (33.8)	0.507
Pulmonary disease, n (%)	11 (13.1)	11 (16.2)	0.393
ASA score, n (%)			0.433
1	12 (14.3)	7 (10.3)	
2	43 (51.2)	42 (61.8)	
3	29 (34.5)	19 (27.9)	

**Table 2 healthcare-10-00531-t002:** Surgical and anaesthesiology parameters.

Variable	AirSeal (N = 84)	Standard (N = 68)	*p* Value
Estimated blood loss, mL			0.880
median (range)	122.0 (77.6–166.4)	121.0 (79.6–162.4)	
SLN detection, n (%)			0.176
Bilateral	70 (83.3)	52 (76.5)	
Monolateral	9 (10.7)	13 (19.1)	
Failed mapping	5 (6)	3 (4.4)	
Lymph nodes removed	4 (0–9)	7 (0–17)	0.101
median (range)			
CO2 IAP, mmHg			
median (range)	8.5 (7.5–9.5)	11.3 (10.2–12.4)	<0.0001
Global pain at 4 h,			
median (range)	1.4 (0.4–2.4)	1.7 (0.9–2.5)	0.023
Global pain at 8 h,			
median (range)	0.8 (0–1.6)	1.4 (0.7–2.1)	<0.0001
Global pain at 24 h,	0.2 (0–0.8)		
median (range)		1.1 (0.2–2.0)	<0.0001
Shoulder pain, n (%)			
Yes	6 (7.1)	20 (29.4)	
No	78 (92.9)	48 (70.6)	<0.0001
Morphine consumption, n (%)			
Yes	4 (4.8)	19 (27.9)	
No	80 (95.2)	49 (72.1)	<0.0001
ETCO2, mmHg,			
median (range)	33.7 (31.4–36.0)	35.8 (32.3–39.3)	<0.0001
Peak airway pressure, cm H_2_O, median (range)			
	21.8 (16.3–27.3)	24.8 (21.5–28.1)	<0.0001
Max systolic arterial pressure, mmHg, median (range)			
	111.7 (98.2–125.2)	133.5 (119.7–147.3)	<0.0001
Total CO2 used, liters,			
median (range)	34.1 (29.9–38.3)	47.9 (41.7–54.1)	<0.0001
Duration of surgery, minutes, median (range)			.
	113 (91.8–134.2)	119 (84.4–153.6)	0.445
Length of stay, days,	2.0 (1.4–2.6)	2.2 (1.4–3.0)	0.224
median (range)			
Major complications, n (%)	0	2 (2.9)	-

**Table 3 healthcare-10-00531-t003:** Analysis in the subgroup of obese women with BMI greater than 30 kg/m2 (N = 77).

Variables	AirSeal	Standard	*p* Value
	(n° = 32)	(n° = 35)	
Age			
median (range)	62 (58–69.3)	65 (57–77)	0.343
Previous surgery			
yes	11 (34.4%)	18 (51.4%)	0.123
Smoke habit			
yes	12 (37.5%)	10 (28.6%)	0.603
Pulmonary disease			
yes	6 (18.8%)	8 (22.9%)	0.457
ASA, n (%)			
1	5 (15.6%)	1 (2.9%)	
2	15 (46.9%)	23 (65.7%)	0.137
3	12 (37.5%)	11 (31.4%)	
Et-CO2,			
median (range)	32 (31–33)	34 (32.4–37)	0.0004
Peak airway pressure, median (range)			
	19 (17–22.4)	25 (20–25)	0.0030
Max systolic pressure			
median (range)	105 (95–116.3)	130 (120–137.2)	<0.0001
Total CO2 used, liters			
Median (range)	30 (30–35)	45 (40–50)	<0.0001
Surgical time, minutes,			
median (range)	100 (90.120)	101.9 (80–130)	0.283
Blood loss, mL,			
median (range)	100 (99.6–100)	100 (100–120)	0.497
NRS pain at 4 h, median (range)			
	1 (0–1)	1 (1–2)	0.062
NRS pain at 8 h			
median (range)	0 (0–1)	1 (0–1)	0.0034
NRS pain at 24 h			
median (range)	0 (0–0)	1 (0–1)	0.0005
Shoulder pain, n (%)			
yes	2 (6.3)	10 (28.6)	0.018
Morphine consumption, n (%)			
yes			
	2 (6.3)	14 (40.0)	0.001
Length of stay, days			
1–2	28 (87.5)	24 (68.6)	
3	4 (12.5)	11 (31.4)	0.058

## Data Availability

The database including study data has a digital excel format, has been anonymized for the analysis and will be available and can be shared in case of need.

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
