# Peer review of "Low-Pressure Laparoscopy Using the AirSeal System versus Standard Insufflation in Early-Stage Endometrial Cancer: A Multicenter, Retrospective Study (ARIEL Study)"

_healthcare, 2022, doi:10.3390/healthcare10030531_

Round 1
Reviewer 1 Report
We congratulate the authors for their work. We would like to ask them some questions:
1- How can they discuss the bias of multiple different surgeons for the procedure?
2- Did they use specific pain control protocols intra op, or post op or ERAS protocols?
Author Response
1- How can they discuss the bias of multiple different surgeons for the procedure?
Response: thank you for the question. All cases were recruited in high volume centers and all surgeon have a well known experience in treating endometrial cases and good skills with MIS. Furthermore, main surgical variables such as blood loss and surgical time particularly in obese patients with BMI greater than 30, or complications were not significant.
2- Did they use specific pain control protocols intra op, or post op or ERAS protocols?
Yes all center shared a similar pain control protocol, and ERAS was already present il all center.
Reviewer 2 Report
Low Pressure Laparoscopy Using the AirSeal System veRsus 2 Standard Insufflation in Early-Stage Endometrial Cancer: A 3 muLticenter, Retrospective Study (ARIEL Study)
Alessandro Buda 1,*, Giampaolo Di Martino 2 , Martina Borghese 3 , Stefano Restaino 4 ,Alessandra Surace 1 , 5 Andrea Puppo 3 , Sara Paracchini 1 , Debora Ferrari 2 , Stefania Perotto 1 , Antonia Novelli 3 , Elena De Ponti 5 , 6 Chiara Borghi 1 , Francesco Fanfani 6,7 and Robert Fruscio
*Comments to the author
In this manuscript, the authors evaluated the benefits of a low-pressure insufflation system (AirSeal) vs standard insufflation system in terms of anesthesiologists’ parameters and postoperative pain in patients undergoing laparoscopic surgery for early-stage endometrial cancer. The authors demonstrated significantly more women in the AirSeal group were discharged on day one compared to the standard group. Results were also confirmed when the subgroup of women with a BMI >30 kg/m. Based on these results authors suggested that the use of low-pressure laparoscopy should be encouraged mainly in obese women with endometrial cancer, also to promote the development of outpatient surgery. This is an interesting piece of work with a direct clinical application. However, I have a minor comment regarding sample size, age group, and race. It is better to include more sample size, a wide range of age groups, and races.
Author Response
Reviewer: "......However, I have a minor comment regarding sample size, age group, and race. It is better to include more sample size, a wide range of age groups, and races".
RESPONSE: thank you for the impressive comment. However, this was a preliminary study to be used as a base for a prospective multicenter staudy that is going to start soon and in our hopes it will include more races with a larger sample size to confirms our preliminary report in women with endometrial cancer.
Reviewer 3 Report
The authors conducted a retrospective study which evaluated the benefits of low-pressure insufflation system (AirSeal) in patients undergoing minimally invasive surgery. They examined the differences of intraoperative anesthesia variables and global pains between AirSeal system and standard one. They concluded that the used of low-pressure laparoscopy has some benefits in minimally invasive therapy for gynecologic diseases and could be a valid alternative to standard laparoscopy, and also promote the development of outpatient surgery. There are some of interests in this article.
I feel that the paper is well written as a pilot study for minimally invasive therapy for gynecologic diseases. And I also think that the AirSeal system has some benefits for laparoscopic surgery from my own experiences. The authors may confirm the benefits scientifically in this paper. If more advanced study such as a randomized controlled trial will be performed, the study design can be improved in the future.
However, it has necessary for minor revision.
- The numbers of figures or tables in the manuscript seem incorrect. The authors should revise the manuscript or figures if needed.
- In table 1, the word of G1 in the item of FIGO stage is out of alignment. The authors should revise that.
Author Response
1. The numbers of figures or tables in the manuscript seem incorrect. The authors should revise the manuscript or figures if needed
RESPONSE: We corrected all the error in the text.
2. In table 1, the word of G1 in the item of FIGO stage is out of alignment. The authors should revise that.
RESPONSE: We corrected all the alignment in the table.